# A microsporidian impairs *Plasmodium falciparum* transmission in *Anopheles arabiensis* mosquitoes

Jeremy K. Herren [1,2✉], Lilian Mbaisi [1,3,8], Enock Mararo [1,8], Edward E. Makhulu [1], Victor A. Mobegi [3,4], Hellen Butungi [1,5], Maria Vittoria Mancini [2], Joseph W. Oundo [1], Evan T. Teal [1], Silvain Pinaud [6], Mara K. N. Lawniczak [6], Jordan Jabara [1], Godfrey Nattoh [1,7] & Steven P. Sinkins [2]

A possible malaria control approach involves the dissemination in mosquitoes of inherited symbiotic microbes to block *Plasmodium* transmission. However, in the *Anopheles gambiae* complex, the primary African vectors of malaria, there are limited reports of inherited symbionts that impair transmission. We show that a vertically transmitted microsporidian symbiont (*Microsporidia MB*) in the *An. gambiae* complex can impair *Plasmodium* transmission. *Microsporidia MB* is present at moderate prevalence in geographically dispersed populations of *An. arabiensis* in Kenya, localized to the mosquito midgut and ovaries, and is not associated with significant reductions in adult host fecundity or survival. Field-collected *Microsporidia MB* infected *An. arabiensis* tested negative for *P. falciparum* gametocytes and, on experimental infection with *P. falciparum*, sporozoites aren't detected in *Microsporidia MB* infected mosquitoes. As a microbe that impairs *Plasmodium* transmission that is non-virulent and vertically transmitted, *Microsporidia MB* could be investigated as a strategy to limit malaria transmission.

[1] International Centre of Insect Physiology and Ecology (ICIPE), Kasarani, Nairobi, Kenya. [2] MRC-University of Glasgow Centre for Virus Research, 464 Bearsden Road, Glasgow G61 1QH, UK. [3] Centre for Biotechnology and Bioinformatics (CEBIB), University of Nairobi, Nairobi, Kenya. [4] Department of Biochemistry, University of Nairobi, Nairobi, Kenya. [5] University of the Witwaterstrand, Wits Research Institute for Malaria, Johannesburg, South Africa. [6] Wellcome Sanger Institute, Wellcome Genome Campus, Hinxton CB10 1SA, UK. [7] Pan African University Institute for Basic Sciences Technology & Innovation, Nairobi, Kenya. [8] These authors contributed equally: Lilian Mbaisi, Enock Mararo. ✉email: jherren@icipe.org

The malaria disease burden remains a major impediment to economic development over many regions of sub-Saharan Africa. Large-scale insecticide treated net (ITN) distribution campaigns over the previous 15 years have reduced malaria cases by an estimated 40%[1]. However, progress has plateaued; between 2014 and 2016 global incidence remained essentially the same[1]. This is a strong indication that current control measures are insufficient and additional novel strategies to control *Anopheles* mosquito populations or their capacity to transmit *Plasmodium* parasites are needed if we are to make further inroads in reducing malaria incidence.

The outcome of vector–pathogen interactions can be influenced by symbiotic microbes. Notably, symbionts can prevent disease vectors from transmitting pathogens that are agents of human disease. This can be developed into a novel vector management strategy; symbionts are disseminated into vector populations to limit their capacity to transmit human disease. For example, it has been demonstrated that *Wolbachia*, which protects mosquitoes from viral pathogens, has great potential for controlling the spread of mosquito-transmitted viral diseases of humans[2–7]. There are reports of *Wolbachia* from anopheline mosquitoes[8,9], in some cases with evidence for a *Plasmodium*-protective phenotype[10,11]. However, there are also indications that *Wolbachia* strains in *An. gambiae s.l.* are either low intensity and have low efficiency of vertical transmission[12], or are artefacts produced by contamination[13]. Being intracellular and maternally inherited, many microsporidians have a similar lifestyle to *Wolbachia*. These obligately intracellular simple eukaryotes, classified within or as a sister group to fungi[14], are known to be harbored by mosquitoes and could be equally useful for the control of vector-borne diseases.

Microsporidia have lifecycles that include a meront phase during proliferation, and spores with chitinous cell walls involved in host to host transmission through spore ingestion. Species with solely horizontal transmission usually show greater virulence and lower host specificity, but where a combination of horizontal and vertical transovarial transmission occurs, lower virulence is advantageous, and is normally associated with a higher degree of host specificity[15]. Sex ratio distortion toward females has been reported (a manipulation characteristic of transovarially transmitted symbionts), for example in the *Dictyocoela* microsporidian of Amphipod crustaceans[16]. Various Microsporidia species have been reported in mosquitoes[17–26], with simple or complex lifecycles[19] but all of which are pathogens where virulence is primarily associated with larval mortality or reduced adult fecundity and lifespan[18–23]. Here, we characterize an apparently non-pathogenic microsporidian from field populations of *An. arabiensis* in Kenya. We link Microsporidia infection to the reduction of *Plasmodium* transmission capacity in this important vector species.

## Results

**Microsporidia detected in *An. arabiensis* field populations**. A microsporidian, designated *Microsporidia MB*, was found to occur at a high intensity and with low to moderate prevalence (0–9%) in geographically dispersed populations of *An. arabiensis* in Kenya (Supplementary Fig. 1). Phylogenetic analysis of the *18S* ribosomal gene revealed that *Microsporidia MB* is related to *Crispospora chironomi*[27], a species recently identified from non-biting midges. The *18S* gene sequence of *Microsporidia MB* shows 97% similarity with *Crispospora chironomi*. *Microsporidia MB* and *Crispospora chironomi* are in clade IV that unites Microsporidia of terrestrial origin infecting diverse hosts[28] (Fig. 1a and Supplementary Fig. 2). The previous reports of microsporidians infecting *Anopheles* mosquitoes all belong to different clades of

Microsporidia[28]. The morphology of *Microsporidia MB* closely resembles *Crispospora chironomi*, exhibiting both polysporoblastic and diplosporoblastic sporogenies, both found in the larval mosquito gut epithelium (Fig. 1b).

**Seasonal variation in *Microsporidia MB* prevalence**. In Kenya, there is a bimodal precipitation pattern, with long rains generally occurring from April to June and short rains between November and December. By screening field-collected *An. arabiensis* mosquitoes we established that *Microsporidia MB* prevalence fluctuates seasonally, with the greatest proportion of infected *An. arabiensis* being observed in January and June, 4–6 weeks after peak rainfall in the Mwea collection site (Fig. 2). At another collection site, Ahero, *Microsporidia MB* prevalence also increased after the rains; however, the timing of this rise was less distinct. Notably, this region has less clearly defined periods of peak rainfall (Fig. 2).

***Microsporidia MB* is vertically transmitted**. *Microsporidia MB* are maternally transmitted with high efficiency, from 45 to 100% (Fig. 3a). The intensity of *Microsporidia MB* in $G_1$ offspring is positively correlated to maternal $G_0$ intensity (Fig. 3b). *Microsporidia MB* were observed in the mosquito ovaries, where the symbiont colonizes and penetrates oocytes (Fig. 3c). We found no evidence that *Microsporidia MB* manipulates the sex ratio of *An. arabiensis* (Fig. 3d).

***Microsporidia MB* impairs *Plasmodium* transmission**. The effects of *Microsporidia MB* on *Plasmodium* infection in *An. arabiensis* was further examined around Mbita point, western Kenya, using $G_1$ offspring pools obtained from field-collected $G_0$ mosquitoes and direct membrane feeding assays (DMFAs) carried out with *Plasmodium falciparum* infected donor blood. Since there was a low to moderate prevalence of *Microsporidia MB* in field populations (Fig. 4, Supplementary Fig. 1), we screened mosquitoes for *Microsporidia MB* and sorted them to ensure that $G_1$ progeny pools used for DMFAs would have both *Microsporidia MB* infected and uninfected mosquitoes. *Plasmodium* was quantified by qPCR 10 days after DMFAs, and a strong negative correlation was apparent between *Microsporidia MB* and the presence of *Plasmodium* in whole mosquitoes (Supplementary Fig. 3).

*Plasmodium* parasites in an ingested bloodmeal undergo a series of developmental changes prior to traversing the peritrophic matrix and midgut epithelium to form a sporogonic oocyst, which releases sporozoites into the hemocoel. From the hemocoel, sporozoites travel to the salivary gland, traverse an epithelium and mix with *Anopheles* saliva resulting in an infectious mosquito, usually 8–14 days after the blood meal[29]. To investigate the stage at which *Plasmodium* development was impaired, we specifically quantified *Plasmodium* in the *An. arabiensis* head and thorax and abdominal compartments. The absence of *Plasmodium* in the head and thorax compartment of *Microsporidia MB* infected *An. arabiensis* 10 days after DMFA indicates that *Microsporidia MB* prevents *An. arabiensis* salivary glands from being colonized by *Plasmodium* sporozoites (Fig. 4a, b) and therefore prevents *Plasmodium* transmission. In addition, *Plasmodium* was not detected by qPCR in the abdomens of *Microsporidia MB* infected *An. arabiensis* 10 days after DMFA (Fig. 4c, d). The absence of *Plasmodium* detected by qPCR in the abdomens of *Microsporidia MB* infected *An. arabiensis* indicates a strong reduction in the development of *Plasmodium* oocysts in the mosquito midgut[30]. To determine to what extent the qPCR assay was able to accurately identify the absence of *Plasmodium* oocysts we carried out both qPCR and microscopic examination

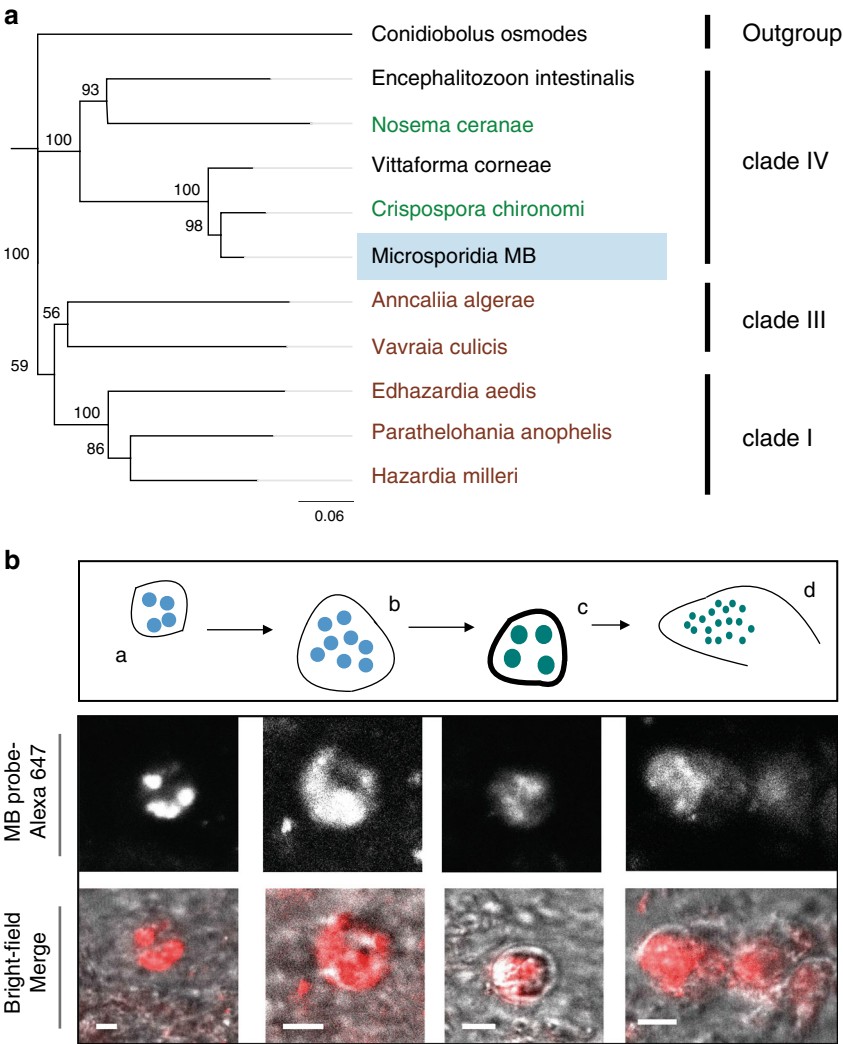

**Fig. 1 A microsporidian associated with *An. arabiensis* populations in Kenya. a** *18S* rDNA-based phylogeny reveals that *Microsporidia MB* are in clade IV of the Microsporidia. Labeled in brown are the microsporidian species known from mosquitoes. In green are microsporidians associated with other insect groups. *Microsporidia MB 18S* rDNA gene sequence deposited in GenBank (accession number MT160806[https://www.ncbi.nlm.nih.gov/nuccore/MT160806]) and in Source Data file. **b** Fluorescence in situ hybridization (FISH) staining of the diplokaryotic stages of *Microsporidia MB* merogony (**a, b**) and spore capsule formation (**c, d**) in *An. arabiensis* larval gut epithelial tissues. Red in the merge images is the MB probe-Alexa 647 FISH probe targeting *Microsporidia MB*. The scale bar is 1.5 μm. Representative images as observed in four independent experiments.

on the same 75 *Anopheles* midguts (Supplementary Table 1). Notably, we observed greater sensitivity in the qPCR-based assay, which is in concordance with other PCR-based studies that observed higher sensitivity for *Plasmodium* DNA detection than other methods[31,32]. The qPCR-based assay detected *Plasmodium* in 26 midguts, whereas only 21 midguts were found to be positive based on microscopy. We note that 2 midguts were found to have a low number of oocysts by microscopy (1,3) and were not found to have *Plasmodium* based on the qPCR assay. This finding suggests that either, (a) qPCR was not able to detect some low-density midgut oocyst infections or, (b) as noted by others[33], there is potential for oocyst misidentification by microscopy in a field laboratory setting. Altogether, these results indicate that *Microsporidia MB* reduces the establishment of *Plasmodium* oocysts in the *Anopheles* midguts and impairs the colonization of *Anopheles* salivary glands by *Plasmodium* sporozoites.

**Microsporidia MB does not confer a fitness cost.** The fecundity and egg to adult survival rate of *Microsporidia MB* infected and uninfected iso-female lineages were examined. No significant

differences were observed in the number of eggs laid by *Microsporidia MB* infected versus uninfected individuals, indicating that unlike most other mosquito microsporidians[17], *Microsporidia MB* does not have a sterilizing effect on females (Fig. 5a). A shortened development period from egg to adult was observed for individuals carrying *Microsporidia MB* (Fig. 5b, c). We investigated the survival of adult female mosquitoes harboring *Microsporidia MB* and found their longevity was similar to uninfected mosquitoes (Fig. 5d). The intensity of *Microsporidia MB* was examined across the lifecycle of the mosquito and was higher in adults than in larvae, but larval intensity is highly variable. Notably, *Microsporidia MB* levels are lower in recently emerged adults, increasing with mosquito age (Fig. 5e).

**Microsporidia MB effects gene expression in An. arabiensis.** To gain insights into the mechanistic basis of the *Microsporidia MB Plasmodium* protection, RNA sequencing was carried out in samples of pooled *An. arabiensis* midguts harboring the *Microsporidia MB* symbiont and uninfected controls (Supplementary Table 2). In addition, RNA sequencing was carried out in pooled

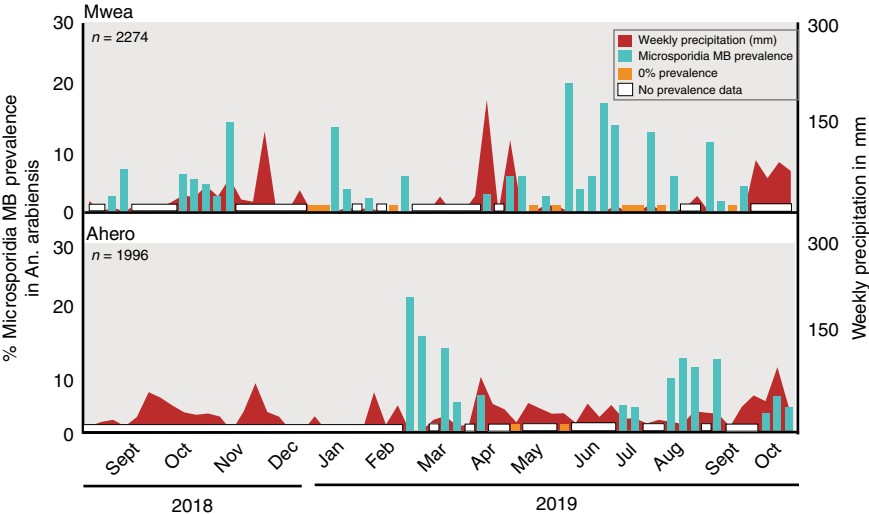

**Fig. 2 Seasonal variation in *Microsporidia MB* prevalence in two sampling sites.** Weekly precipitation values given are based on earth observation and are shown in dark blue. Light blue bars indicate the prevalence of *Microsporidia MB*, monitored in weekly collections. Overall, 136 out of 2274 (6%) female *An. arabiensis* were found to be infected with *Microsporidia MB* in Mwea and 159 out of 1996 (8%) female *An. arabiensis* in Ahero. Source data are provided as a Source Data file.

samples of *An. arabiensis* ovaries, since these tissues were found to have a high intensity of *Microsporidia MB* infection (Supplementary Table 3). Full-length RNA sequencing (Oxford Nanopore Technologies) was used to identify candidate genes involved in impairing *Plasmodium* transmission. Out of 13,166 transcripts from the *An. arabiensis* (AARA) transcriptome, 8611 transcripts were detected with ≥10 reads. We found that genes transcribed at higher levels in *An. arabiensis* midguts in the presence of *Microsporidia MB* compared to uninfected midguts were predominately involved in digestion and immunity. In the midgut the most upregulated genes in the presence of *Microsporidia MB* were the serine proteases *ISP13* (AARA001672-RA), *antryp-4* (AARA016538-RA) and *antryp-1* (AARA016534-RA), the female reproductive tract protease *GLEANR_896* (AARA009400-RA), and the anti-microbial peptides *cecropin-2* (AARA015679-RA) and *gambicin* (AARA010954-RA). In the ovaries, *Microsporidia MB* was associated with high levels of *c-type lysozyme* (AARA016504-RA), *transferrin* (AARA010742-RA) and Salivary gland (AARA001017-RA, AARA008387-RA) and Salivary gland-like protein (AARA016177-RA) genes.

## Discussion

Several *Anopheles*-associated microsporidians have been shown to interfere with the infection and development of *Plasmodium*[18,34,35]. *Nosema stegomyiae* disrupts the development of the oocysts in *An. gambiae*, attributed to midgut degradation and consequent disruption of *Plasmodium* binding[18], while *Vavraia culicis* impairment of development of *Plasmodium* has been associated with host innate immune priming[35]. The impairment observed in previous *Plasmodium* transmission experiments have been relatively modest, with only partial reduction of transmission observed. In applied terms, given their virulence, their primary application would be as population suppression agents and the *Plasmodium* transmission impairment would provide a minor add-on. In contrast, high levels of impairment of *P. falciparum* transmission were observed here for *Microsporidia MB*. Additionally, *Microsporidia MB* interferes with *Plasmodium* development early, prior to the formation of *Plasmodium* oocysts in the *Anopheline* mosquito gut.

*Microsporidia MB* infected female *An. arabiensis* transmit the infection with high efficiency to their offspring. *Microsporidia MB*

were observed in *An. arabiensis* ovaries suggesting a transovarial transmission route. It is notable that a number of females with low *Microsporidia MB* intensity did not transmit or transmitted very poorly to their offspring. It is possible that these infections might be newly horizontally acquired and have not yet become localized to the ovaries, a likely requirement for high-efficiency maternal transmission. The identification of *Microsporidia MB* spores in the midgut is suggestive of a horizontal transmission route, which is most likely to take place in larval habitats or potentially during mosquito mating. Environmental conditions appear to influence *Microsporidia MB* infection prevalence in the field, because the highest infection rates were generally observed 4–6 weeks after peak rainfall. It is conceivable that the rates of vertical, and possibly horizontal, transmission are influenced by environmental factors.

*Microsporidia MB* infection does not have an overt effect on *An. arabiensis* fitness, as females harboring *Microsporidia MB* laid an equivalent number of eggs and had a similar lifespan to uninfected controls. Vertically transmitted symbionts are generally selected to minimize fitness impacts on their hosts[36] and this could explain *Microsporidia MB* levels being lower in larvae and increasing modestly across the life of adult *An. arabiensis*. The fitness of adult mosquitoes is highly correlated to their size at eclosion[37] and maintaining low *Microsporidia MB* infection levels in larvae could be important to lessen potential impacts on larval growth rate and hence minimize overall host fitness cost. *An. arabiensis* infected with *Microsporidia MB* were found to have a slightly faster larval development period. It is notable that another microsporidian *Vavraia culicis*, also shortened mosquito development time in *Culex pipiens* but in contrast to *Microsporidia MB* this was associated with a high fitness cost[38]. *Microsporidia MB* could also be affecting host metabolic processes or nutrient availability leading to more rapid host growth, as has been suggested for other vertically transmitted symbionts that have mutualistic phenotypes in *An. gambiae*[39].

Elucidating the mechanistic basis of endosymbiont-induced protective phenotypes presents a number of challenges. In systems where these mechanisms have been investigated, there are indications that multiple processes may be involved[40]. To gain insight into the *Plasmodium*-protective phenotype associated with *Microsporidia MB*, we investigated changes in host gene

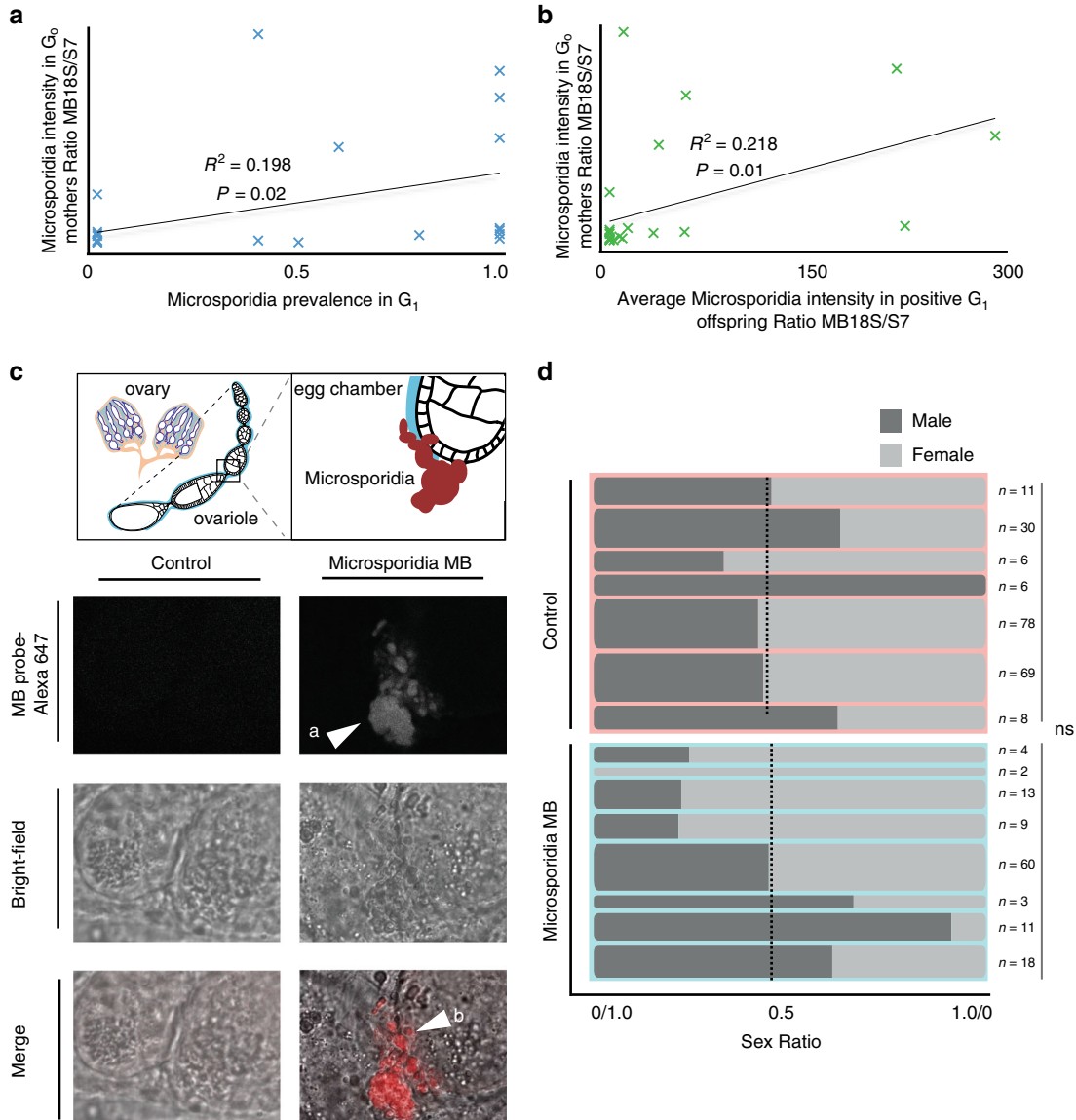

**Fig. 3 *Microsporidia MB* is maternally transmitted and does not bias sex ratio. a** The vertical transmission efficiency of *Microsporidia MB* to $G_1$ depends on the $G_0$ (maternal) *Microsporidia MB* intensity. In a linear regression, the slope is significantly non-zero ($P = 0.02$, $F = 6.169$, and $n = 27$). **b** The intensity of *Microsporidia MB* in $G_1$ offspring is correlated to $G_0$ maternal intensity, with a linear regression the slope that is significantly non-zero ($P = 0.01$, $F = 6.98$ and $n = 27$). **c** Fluorescence microscopy with a *Microsporidia MB* specific FISH probe (MB probe) indicates that *Microsporidia MB* (**a**) is localized to the posterior of developing vitellogenic egg chambers (**b**) in *An. arabiensis*. Scale bar, 20 μm. Representative images as observed in three independent experiments. **d** The sex ratio of broods from *Microsporidia MB* infected mothers does not differ significantly from *Microsporidia MB* uninfected mothers suggesting there is no sex ratio distortion (Unpaired two-tailed *t*-test, $P = 0.32$, $t = 1.02$, df=13). Dashed line reflects overall sex ratio (Uninfected control = 100:108 or 52% female, *Microsporidia MB* = 56:64 or 54% female). Column widths reflect sample sizes Source data are provided as a Source Data file.

expression. *Microsporidia MB* apparently increased the expression of a number of genes, which are involved in digestion (*antryp-4*, *antryp-1*) and immunity (*ISP13*, *cecropin-2* and *gambicin*)[41,42]. The genes *ISP13* and *cecropin-2* have been shown to be upregulated in response to infection[40,43] and *gambicin* has been shown to be an anti-*Plasmodium* factor in *An. gambiae*[44], which lends support to the hypothesis that the impairment of *Plasmodium* transmission by *Microsporidia MB* could be linked to immune priming. *Antryp-4* is a digestive serine protease that is constitutively expressed in unfed *An. gambiae* mosquitoes, whereas *antryp-1* is upregulated upon blood-feeding[41]. The finding that *antryp-4* and *antryp-1* are upregulated in the presence of *Microsporidia MB* suggests that this microsporidian

could affect host metabolism, which may also have implications for vector capacity[45–47].

These findings are significant in terms of regional malaria transmission and epidemiology as well as risk-mapping, and particularly in terms of the development of microbe-based *Plasmodium* transmission-blocking tools. *Microsporidia MB* can be added to the *Anopheles*-associated gut bacteria that have previously been identified as able to reduce *Plasmodium* transmission when introduced into the mosquito[48,49], or following introduction of anti-*Plasmodium* transgenes[50]. In addition, the impairment of *Plasmodium* transmission combined with the key characteristics that will facilitate artificially elevating its population frequency, namely spore production that is likely to facilitate

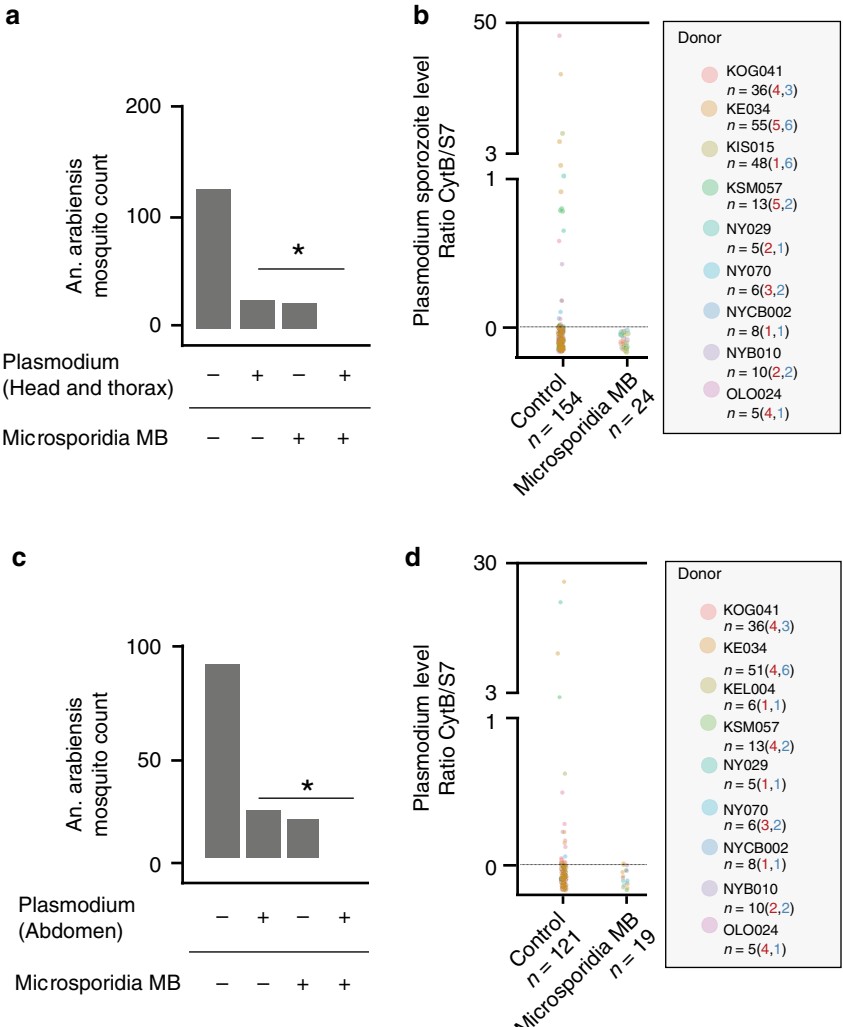

**Fig. 4 *Microsporidia MB* impairs parasite development in *An. arabiensis* after direct membrane feeding assay challenge with *P. falciparum*.** The *Plasmodium* infection rate in *Microsporidia MB* positive and *Microsporidia MB* negative mosquitoes was determined by qPCR. **a** The head and thorax *Plasmodium* infection rate, reflecting presence of sporozoites, in *Microsporidia MB* positive and *Microsporidia MB* negative mosquitoes. There was a significant absence of co-infected mosquitoes (two-tailed fisher exact test, $P = 0.02$ $n = 178$). **b** *Plasmodium* intensity in *An. arabiensis* heads and thoraxes, quantified by qPCR. **c** The abdomen *Plasmodium* infection rate, reflecting presence of oocysts, in *Microsporidia MB* positive and *Microsporidia MB* negative mosquitoes. There was a significant absence of co-infected mosquitoes (two-tailed fisher exact test, $P = 0.04$, $n = 140$). **d** *Plasmodium* intensity in *An. arabiensis* abdomens, quantified by qPCR. Data shown in (**a**, **b**, **c**, **d**) are pooled from replicate experiments carried out using different gametocyte donors (for each donor the numbers of *Plasmodium* positive *An. arabiensis* is shown in red and *Microsporidia MB* positives An. arabiensis is shown in blue). Each data point is an individual An. arabiensis mosquito. Source data are provided as a Source Data file.

dissemination, efficient transovarial transmission and apparently non-virulent interactions with *An. arabiensis* mosquitoes make *Microsporidia MB* is a realistic candidate for a stable vector population replacement strategy. As an unmodified, *Anopheles*-associated inherited endosymbiont, it provides attractive prospects for malaria control.

## Methods

**Sampling sites and collection**. *Microsporidia MB* and *Plasmodium* prevalence in wild *An. arabiensis* mosquitoes was determined by collecting adult female mosquitoes from sites around Kenya: Mbita (Nyawiya, Mageta and Kirindo), Mwea (Mbui-Njeru), Busia (Funyala) and Ahero (Kigoche). *An. arabiensis* were collected inside houses and sheds using CDC light traps and by manual aspiration. For the establishment of *Microsporidia MB* harboring lines gravid mosquitoes were collected solely by manual aspiration inside houses and sheds. All mosquitoes were transported from the field to the *icipe*-Thomas Odhiambo Campus (*iTOC*) laboratories and insectaries alive in cages, each sample represents an individual wild caught mosquito.

**Mosquito species identification**. All experiments were carried out on wild collected *Anopheles gambiae sl.*, which were identified morphologically. In all of the collection sites, *An. arabiensis* is the most common member of the *An. gambiae* species complex, with >97% of complex members being identified as *An. arabiensis*. The species designation was confirmed using a molecular assay that differentiates *An. gambiae s.s.* and *An. arabiensis* using the SINE S200 X6.1 locus[51]. *Anopheles* samples that were not confirmed to be *An. arabiensis* were excluded from analysis.

**Seasonal fluctuations and climate data**. *An. arabiensis* were collected by aspiration in houses on a weekly basis. Specimens were transported to laboratories in Nairobi and Mbita for species identification and *Microsporidia MB* screening. Rainfall data for Mwea (−37.3538W,−0.6577N) and Ahero (−34.9190W, −0.1661N) were obtained from Climate Engine[52], Desert Research Institute and University of Idaho, accessed on (01/12/2019), http://climateengine.org.

**Determination of molecular phylogeny of *Microsporidia MB***. *Microsporidia MB* positive *An. arabiensis* were initially identified by sequencing *18S* amplicons amplified by the SSU rRNA primer pair F: 5′-CACCAGGTTGATTCTGCC-3′; R: 5′-TTATGATCCTGCTAATGGTTC-3′, which targets phylogenetically diverse microsporidians[53]. The primer pair RPOBMBF 5′-ACAGTAGGTCACTTGAT

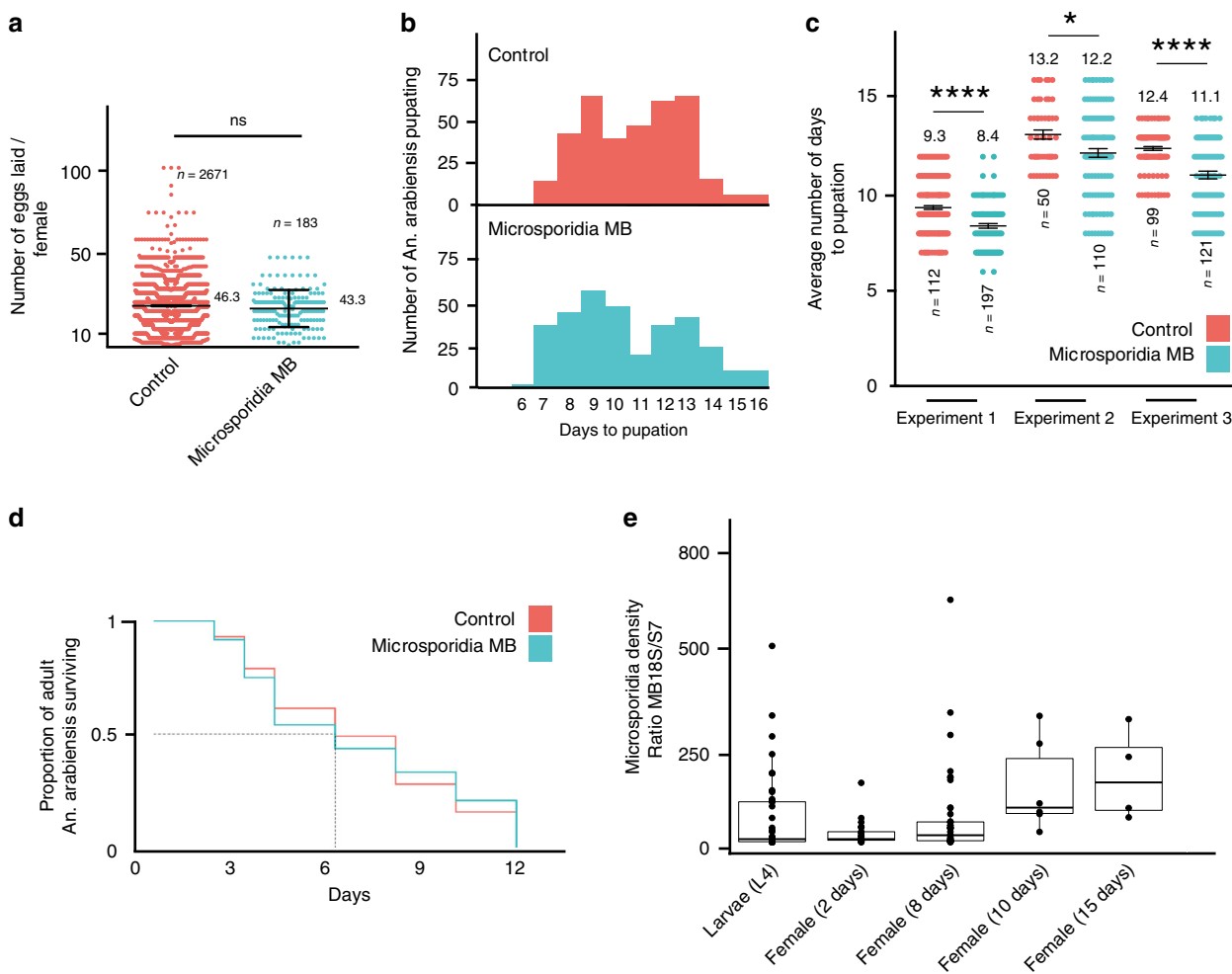

**Fig. 5 *Microsporidia MB* does not overtly decrease host fitness. a** *Microsporidia MB* harboring wild-caught An. arabiensis did not lay significantly less eggs than uninfected counterparts (two-tailed Mann–Whitney test, $P = 0.18$). The black line indicates the mean and error bars reflect SEM. **b** The $G_1$ larval progeny of *Microsporidia MB* infected wild-caught females develop significantly faster than uninfected counterparts. **c** The larval development of *Microsporidia MB* infected An. arabiensis is on average 1.1, 1.4, and 1.3 days less than uninfected controls in three independent experiments (Two-tailed Mann–Whitney test, $P < 0.0001$, $P = 0.03$, $P < 0.0001$, error bars reflect SEM). Mean values are shown above the scatter dot plots. Data shown in (**b**) and (**c**) are from the same three independent experiments. **d** The survival of adult $G_1$ progeny of *Microsporidia MB* infected wild-caught females is not significantly different from uninfected counterparts, one experiment (ii) shown of three independent experiments (i-iii); (i, $P = 0.86$, $n > 9$, ii, $P = 0.97$, $n > 41$, ii, $P = 0.06$ $n > 50$, $n$ denotes the minimum number of mosquitoes per condition). **e** The intensity of *Microsporidia MB* across different developmental stages. The intensity of *Microsporidia MB* is highly variable in larvae, lower in young adults but increases as adult An. arabiensis females age. Boxplot boundaries reflect the inter-quartile range. The horizontal bar is the median and whiskers extend to 1.5 times the inter-quartile range beyond the boxplot. Each datapoint represents an individual mosquito from a total of five (L4-day 2), four (day 8) and three (day 10 and 15) independent experiments. Source data are provided as a Source Data file.

TGAATGTC-3′ and RPOBMBR 5′-TACCATGTGCTTAAGTCTTTGGT-3′ was used to amplify the *rpoB* gene of *Microsporidia MB*. Amplicons were prepared from nine individual *An. arabiensis* from geographically dispersed sites, all individuals had identical *18S* and *rpoB* fragment gene sequences. Prepared amplicons were cleaned using the USB® ExoSAP-IT® PCR Product Cleanup kit according to manufacturer's instructions and sent to Macrogen (Netherlands) for sequencing. Multiple sequence alignment was done using the MUSCLE algorithm[54] version 3.5 in Geneious Prime[55] version 2020.03 (www.geneious.com) alongside reference sequences of other Microsporidia species obtained from NCBI. Tamura-Nei genetic distance model alongside Neighbour-joining tree building algorithm was used in the creation of phylogenies and evaluated with 10000 replicates bootstrap support and 50% support threshold. Rooting was done using *Conidiobolus osmodes* as an outgroup. *Microsporidia MB 18S* rDNA and *rpoB* partial gene reference sequences have been submitted to Genbank, submission ID 2308213. The Accession numbers of sequences used for generating phylogenetic trees are given in Supplementary Table 4.

**Egg laying and establishment of lines**. Wild-caught gravid female mosquitoes were induced to oviposit inside a perforated 1.5 ml micro centrifuge tube containing 50 μl of distilled water and a soaked piece of Whatman paper towel (1 cm × 1 cm in size). Eggs from each female were counted under a compound microscope using a paint brush and then dispensed into water tubs for larval development under optimal rearing conditions (a temperature of 30.5 °C and 30% humidity). Upon laying eggs, the $G_0$ females were screened for presence of *Microsporidia MB* by PCR. The progeny from *Microsporidia MB* positive $G_0$ females were used for: (i) the quantification of vertical transmission efficiency and (ii) the establishment of iso-female *Microsporidia MB* infected lines, whereas lines with similar L1 larvae numbers (±10 individuals) were used as matched negative controls for quantifying development time and survival. For each iso-female line two $G_1$ L4 larvae were screened to determine species and confirm infection status and the line was only considered infected if both were *Microsporidia MB* positive. For the quantification of intensity across developmental stages infected iso-female lines were established. We also scored the sex ratio of progeny by allowing specimens to eclose and then counting males and females. To establish mixed (*Microsporidia MB* positive and negative) *An. arabiensis* pools for direct membrane feeding assays, eggs from *Microsporidia MB* infected and uninfected females were combined and reared together at an approximate ratio of 3:1 (*Microsporidia MB* infected: *Microsporidia MB* uninfected). Infection status was not determined at the larval stage for mixed pools.

**Mosquito rearing**. Larvae were reared under a controlled environment at 30.5 °C (±2 °C). Larvae were fed daily on TetraMin™ baby fish food and fresh double-distilled water added into their tubs every other day to maintain oxygen levels. Adult mosquitoes were reared at 30 °C (±2 °C) and 70% humidity with a constant 12-h day/night cycle. The adult mosquitoes were fed on 6% glucose soaked in cotton wool. For adult survival and larval development time measurements, the status of larvae and mosquitoes was recorded every 24 h.

**Plasmodium direct membrane feeding assays**. *Plasmodium* screening of human subjects was done in the regions surrounding Mbita using Rapid Diagnostic Test (RDT) kits (SD Bioline, UK). Microscopy was carried out on RDT-positive samples to confirm the presence of *P. falciparum* gametocytes. Gametocyte-positive blood used was mixed with an anticoagulant (heparin) and a total volume of 500 µl was placed into mosquito mini-feeders at 37 °C and covered in stretched parafilm. Mixed pools of 2–3-day-old *An. arabiensis* (containing co-reared *Microsporidia MB* positive and negative mosquitoes) were starved for 5 h (the sucrose source was replaced with water for the first 4 h of starvation) prior to the direct membrane feeding assay. Mosquitoes were allowed to feed for an hour after which non blood-fed individuals were discarded. Blood-fed mosquitoes were then maintained for a period of 10 days post-infection and processed for the molecular detection of *Microsporidia MB* and *Plasmodium* oocysts and sporozoites. We only include experiments where there was greater than zero prevalence of *Microsporidia MB* and *Plasmodium* in the mixed pools of *An. arabiensis*.

**Specimen storage and DNA extraction**. All *An. arabiensis* specimens were dry frozen at −20 °C in individual microcentrifuge tubes prior to DNA extraction. Prior to extraction, mosquitoes were sectioned into head and thorax (for detection and quantification of *Plasmodium* sporozoites and *Microsporidia MB*) and abdomens (for detection and quantification of *Plasmodium* oocysts and *Microsporidia MB*). DNA was extracted from each section individually using the protein precipitation method (Puregene, Qiagen, Netherlands).

**Molecular detection and quantification of Microsporidia MB**. *Microsporidia MB* specific primers (MB18SF: CGCCGGCCGTGAAAAATTTA and MB18SR: CCTTGGACGTGGGAGCTATC) were designed to target the *Microsporidia MB 18S* rRNA gene region and tested for specificity on a variety of Microsporidia-infected mosquito controls (including *Hazardia, Parathelohania* and *Takaokaspora*). For detection, the PCR reaction volume was 10 µl, consisting of 2 µl HOTFirepol® Blend Master mix Ready-To-Load (Solis Biodyne, Estonia, mix composition: 7.5 mM Magnesium chloride, 2 mM of each dNTPs, HOT FIREPol® DNA polymerase), 0.5 µl of 5 pmol µl⁻¹ of both forward and reverse primers, 2 µl of the template and 5 µl nuclease-free PCR water. The PCR cyclic conditions used were; initial denaturation at 95 °C for 15 min, further denaturation at 95 °C for 1 min, followed by annealing at 62 °C for 90 s and extension at 72 °C for a further 60 s, all done for 35 cycles. Final elongation was done at 72 °C for 5 min. *Microsporidia MB* was also quantified by qPCR using MB18SF/ MB18SR primers, with normalization against the *Anopheles* ribosomal *S7* host gene[56] (primers, S7F: TCCTGGAGCTGGAGATGAAC and S7R: GACGGGTCTGTACCTTCTGG). The primer efficiencies were determined using published methods[57]. Samples were considered negative if the cycle threshold (Ct) value was greater than 35 or if the melt curve did not align with the positive control. These conditions were met for all negative control samples, which was DNA extracted from *An. arabiensis* from the insectary at icipe, Nairobi. The qPCR reaction volume was 10 µl, consisting of 2 µl HOT FIREPol® EvaGreen® HRM no ROX Mix (Solis Biodyne, Estonia, mix composition: 12.5 mM Magnesium chloride, EvaGreen® dye, BSA, dNTPs, HOT FIREPol® DNA Polymerase and 5× EvaGreen® HRM buffer), 0.5 µl of 5 pmol µl⁻¹ of both forward and reverse primers, 2 µl of the template and 5 µl nuclease-free PCR water. Finally, melt curves were generated including temperature ranges from 65 °C to 95 °C. Standard curves were also generated to determine amplification efficiency. *An. arabiensis* were considered to be infected if *Microsporidia MB* was detected in either head and thorax or abdomen compartments. All reactions were carried out on a MIC qPCR cycler (Bio Molecular Systems, Australia).

**Molecular detection and quantification of Plasmodium**. A qPCR-based assay was used to detect and quantify the *cytochrome b* gene of *Plasmodium*[58]. When used on *An. arabiensis* head and thorax DNA samples 10 days post DMFA this assay can be used to detect and quantify *Plasmodium* sporozoites. When used on *An. arabiensis* abdomen DNA samples 10 days post DMFA this assay can be used to detect and quantify *Plasmodium* oocysts[30]. To confirm that the qPCR-based assay was able to detect oocysts in a manner comparable to microscopy, midguts were dissected, stained and oocysts counted prior to them being removed from slides for DNA extraction and qPCR[59] (Supplementary Fig. 4). *Plasmodium cytochrome b* was normalized against the *Anopheles* ribosomal *S7* host gene[56]. Samples were considered negative if the cycle threshold (Ct) value was greater than 35 or if the melt curve did not align with the positive control. These conditions that were met for all negative control samples, which was DNA extracted from *An. arabiensis* from the insectary at icipe iTOC, Mbita, fed on heat-inactivated Gametocyte-positive blood 10 days prior to extraction. The qPCR mastermix was composed of 2 µl HOT FIREPol® EvaGreen® HRM no ROX Mix (Solis Biodyne,

Estonia, mix composition: 12.5 mM Magnesium chloride, EvaGreen® dye, BSA, dNTPs, HOT FIREPol® DNA Polymerase and 5× EvaGreen® HRM buffer), 0.5 µl of 5 pmol µl⁻¹ of both forward (cytBF) and reverse (cytBR) primers, 2 µl of the template and 5 µl nuclease-free PCR water. The PCR profile for target gene amplification included an initial denaturation at 95 °C for 15 min, further denaturation at 95 °C for 30 s, followed by annealing at 60 °C for 45 s and extension at 72 °C for 45 s repeated for 40 cycles. Final elongation was performed at 72 °C for 7 min followed by generation of a melt curve ranging from 65 °C to 95 °C. A standard curve was generated to determine the PCR efficiency. All reactions were carried out on a MIC qPCR cycler (Bio Molecular Systems, Australia).

**Fluorescence in situ hybridization (FISH)**. DNA probes specific to *Microsporidia MB 18S* rDNA were designed and synthesized (5′-CY5-CCCTGTCCACTA-TACCTAATGAACAT-3′, Macrogen, Netherlands). FISH was conducted on *An. arabiensis* adult and larval tissue specimens using a previously described protocol[60], with minor modifications. Briefly, mosquito tissues (larval gut and adult ovaries) were fixed in 4% Paraformaldehyde (PFA) solution overnight at 4 °C and subsequently transferred into 10% hydrogen peroxide in 6% alcohol for 3 days prior to rehydration in Phosphate Buffer Saline with Tween-20 (PBS-T) for 1–2 h. Hybridization was conducted by incubating tissues in 150 µl hybridization buffer (20 mM Tris-HCl [pH 8.0], 0.9 M NaCl, 0.01% sodium dodecyl sulfate, 30% formamide) containing 100 pmol ml⁻¹ of the probe at room temperature overnight. After washing with PBS-T, the hybridized samples were placed on a slide and were visualized immediately using a Leica SP5 confocal microscope (Leica Microsystems, USA). Images were analyzed with the ImageJ 1.50i software package[61].

**RNA sequencing**. Two pools of tissue samples from *Microsporidia MB* infected *An. arabiensis* (3 midguts, 12 ovaries) and two pools of non-infected *An. arabiensis* (5 midguts, 3 ovaries) were used for RNA extraction. TRIzol™ reagent (Invitrogen) was used to extract both RNA and DNA from the same *An. arabiensis* pooled midgut samples. RNA was precipitated with isopropanol, then DNA was extracted from remaining organic and interphase by adding TNES back-extraction buffer (BEB) prior to precipitation with isopropanol. After two 75% ethanol cleans, DNA and RNA was resuspended in nuclease-free water and quantified with nanodrop. DNA and RNA quality were assessed by checking the size distribution on an electrophoresis-based Bioanalyzer instrument and high-sensitivity DNA and RNA 6000 Nano kits, respectively. RNA was used for long-read RNA sequencing while DNA was used to confirm *Microsporidia MB* infection by PCR. Extracted RNA was released to the long-read team of the Wellcome Sanger Institute for PCR-cDNA library preparation following manufacturer's instructions (Oxford Nanopore Technologies, UK). 50 ng of total RNA was used for library preparations. Long-reads from ONT were then aligned against the *An. arabiensis* transcriptome (Dongola_AaraD1.11) reference using minimap2 -map-ont. Mapped reads were then used to estimate transcript abundance manually using samtools-1.6 (samtools-1.6 view sorted_bam -f 2 -q 5 | cut -f 3 | sort | uniq -c > ./counts/count.txt). The total number of reads obtained was 3,061,001 for the *Microsporidia MB* infected midgut pool, 195,223 for the uninfected midgut pool, 1,820,596 for the *Microsporidia MB* infected ovary pool and 390,355 for the uninfected ovary pool. Reads from both pools were randomly resampled to until the total number for each pool was equal to the uninfected read count, which was lower, for each tissue type. The genes corresponding to these reads were then ranked in order of abundance in the *Microsporidia MB* infected resampled counts relative to absence in *Microsporidia MB* uninfected resampled counts.

**Statistical analysis**. We carried out statistical analyses using the two-tailed unpaired t-test to compare unpaired data values with a normal distribution. Where data distributions were non-normal, the two-tailed Mann–Whitney u test was used. Linear regression was used to determine correlation coefficients and significance. To analyze the significance of contingency table data, a two-tailed Fisher exact test was used. The Log-Rank test was used to determine levels of statistical significance for survival data. All statistical analyses were preformed using Graphpad Prizm version 6.0c software or R (version 3.5.3). P-values of $*p < 0.05$, $**p < 0.01$, $***p < 0.001$ and $****p < 0.0001$ were deemed to be statistically significant.

**Ethics statement**. Ethical clearance (Kenya Medical Research Institute Scientific and Ethics Review Unit: KEMRI/RES/7/3/1 and Glasgow MLVS College Ethics Committee: Project Number 200170001) was obtained prior to human blood sample collection. Written informed consent was sought from parents and guardians of the children to allow minors to participate in the study. Consent was also obtained from heads of households that provided approval for indoor mosquito collection.

**Reporting summary**. Further information on research design is available in the Nature Research Reporting Summary linked to this article.

## Data availability

All data supporting the findings of this study are available within the article and its Supplementary Information files, or are available from the authors upon request. The

source data used to generate Figs. 2–5 and Supplementary Figs. 1–3 and Supplementary Tables 1–3 is available in the Source Data file. The *Microsporidia MB 18 S* partial gene sequence has been submitted to Genbank with accession number MT160806[https://www.ncbi.nlm.nih.gov/nuccore/MT160806]. The *Anopheles arabiensis* RNA sequencing data have been submitted to the NCBI Sequence Read Archive (SRA) database under accession number PRJNA622655 [https://www.ncbi.nlm.nih.gov/sra/?term=PRJNA622655].

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

## Acknowledgements

The authors acknowledge Milcah Gitau of *icipe* Arthropod Rearing and Containment Unit for mosquito rearing assistance. We thank Ibrahim Kiche, Faith Kyengo, Ulrike Fillinger, Dan Masiga, Jandouwe Villinger and Oscar Mbare for assistance and advice and Quan Lin for assistance releasing RNA sequencing data. This work was supported by the Wellcome Trust [107372, 200274, 202888, 206194], the BBSRC [BB/R005338/1, sub-grant AV/PP015/1], the Scottish Research Council, the Swiss National Science Foundation [P2ELP3_151932], the R. Geigy Foundation, the UK's Department for International Development (DFID); Swedish International Development Cooperation Agency (Sida); the Swiss Agency for Development and Cooperation (SDC); Federal Democratic Republic of Ethiopia and the Kenyan Government.

## Author contributions

J.K.H conceived and designed the majority of the experiments. E.M, L.M., and E.E.M. performed the majority of experiments. J.W.O, E.E.M., J.J., and E.T. collected mosquitoes, screened them and prepared them for experiments. H.B. and M.V.M. contributed to transmission-blocking and molecular identification experiments, respectively. G.N., E.T., and E.E.M prepared samples for RNA sequencing analysis. M.K.N.L and S.P. conducted the RNA sequencing and analysis. J.K.H., S.P.S., E.M., E.E.M., L.M., J.J., M.V.M., V.A.M., M.K.N.L., S.P., and E.T analyzed the data. J.K.H and S.P.S. wrote the manuscript.

## Competing interests

The authors declare no competing interests.
