## [Peer Review File · Nature Communications]

Editorial Note: This manuscript has been previously reviewed at another journal that is not operating a transparent peer review scheme. This document only contains reviewer comments and rebuttal letters for versions considered at Nature Communications .

Reviewers' Comments:

Reviewer #3:

Remarks to the Author:

Herren and co-authors found a new species of Microsporidia that negatively correlate with Plasmodium infection in An arabiensis and that is maternally transmitted. They suggest that this microsporidia blocks Plasmodium infection in the mosquito and that its maternal transmission makes it a new candidate for population replacement. This is further supported by the evidence that there is no obvious fitness cost in microsporidia infected mosquitoes. This potential outcome is promising and therefore may be of interest to the wide readership of Nature communications.

The editor asked me to specifically consider the rebuttal to reviewer 2 in this review. I overall agree with the main points raised by reviewer 2, ie that while this paper may put light on a promising candidate, the current level of characterisation is a bit preliminary. I feel that some of his/her points have not been addressed.

First, there is no data on oocyst counts and the authors do not explain how they set their "zero infection" after qPCR analysis, which makes it a bit more difficult to claim that there is a strict blockade of infection. Some infections with oocyst counts do not appear as a big deal and would strengthen the data.

Second, the question on sex distorsion has not been addressed, and again, the experiment seems straightforward.

Finally, Figure 3A is still not clear, it would make more sense to mention "prevalence" and "intensity" (if that's what the authors mean) and use XY scatters with density in mother as X and infection in the progeny as Y. Also, the authors need to be consistent on the use of G1 or F1 (and the s following G1 is misleading).

I have two other comments:

- The current manuscript leaves the reader with an impression that not many things have been done to characterise the system, while some information placed in the supplementals give a more thorough idea on the extent of the work. They should therefore be placed in the main body (ED Fig 4A (+ add a legend on colours); ED fig 3A,B to be exchanged with Fig2A,B which is too complicated).
- If the authors suggest that blockade occurs before the oocyst stage, it would make sense to check whether melanized ookinetes can be observed in the mosquito guts. This could be done in the same experiment as oocyst counts mentioned above.

Overall, the manuscript would require some polishing. This includes some typos (lines: 28, 31, 73, 101, 267), but also some work required on the text in general. The last sentence of the abstract should be tuned down not to oversell a tool that is not yet developed.

Finishing on a positive note, I would like to point that the authors have used a very relevant mode of infection with Plasmodium: they used the blood of patients, did not replace the sera to be in conditions as close as possible to what mosquitoes would really encounter in the field. Moreover, they used a good number of replicates.

Responses to reviewer comments:

Reviewers' comments:

Reviewer #3 (Remarks to the Author):

Herren and co-authors found a new species of Microsporidia that negatively correlate with Plasmodium infection in *An. arabiensis* and that is maternally transmitted. They suggest that this microsporidia blocks Plasmodium infection in the mosquito and that its maternal transmission makes it a new candidate for population replacement. This is further supported by the evidence that there is no obvious fitness cost in microsporidia infected mosquitoes. This potential outcome is promising and therefore may be of interest to the wide readership of Nature communications.

We thank the reviewer for this assessment of the implications of our findings.

The editor asked me to specifically consider the rebuttal to reviewer 2 in this review. I overall agree with the main points raised by reviewer 2, ie that while this paper may put light on a promising candidate, the current level of characterisation is a bit preliminary. I feel that some of his/her points have not been addressed.

We are confident that we have now addressed the vast majority of the points originally raised by reviewer 2.

First, there is no data on oocyst counts and the authors do not explain how they set their "zero infection" after qPCR analysis, which makes it a bit more difficult to claim that there is a strict blockade of infection. Some infections with oocyst counts do not appear as a big deal and would strengthen the data.

We have carried out a number of MFA specifically to validate our qPCR-based *Plasmodium* assay. We carried out microscopy and then extracted DNA from the same midguts. Despite minor discrepancies, we found good alignment between the positivity and even intensities of infections observed in both cases. Both techniques are known to have advantages and disadvantages and carrying out both method on every sample would have been prohibitively challenging. We hope that having carried out the majority of our experiments by qPCR, but having validated the qPCR with a number of microscopy + qPCR experiments will be sufficient to confirm our data is accurate. We have now noted in the materials and methods that our negative controls (mosquitoes fed on heat inactivated gametocyte-positive blood) were used to set zero infection. All negative controls had CT values over 35. Samples deemed negative also had CTs over 35 or, in rare cases, melting peaks that did not correspond to the controls.

Second, the question on sex distortion has not been addressed, and again, the experiment seems straightforward.

We have now incorporated the sex ratio data for broods from *Microsporidia MB*-infected and uninfected iso-female lines. Overall, these data do not provide evidence for significant sex ratio distortion.

Finally, Figure 3A is still not clear, it would make more sense to mention "prevalence" and "intensity" (if that's what the authors mean) and use XY scatters with density in mother as X and infection in the progeny as Y. Also, the authors need to be consistent on the use of G1 or F1 (and the s following G1 is misleading).

We have replaced the word “density” with “intensity” in this figure and throughout the manuscript for consistency. We agree with the reviewer and have replaced all the rather confusing bar charts with XY scatter plots. Issues with G1 and F1 have now been corrected.

I have two other comments:

- The current manuscript leaves the reader with an impression that not many things have been done to characterise the system, while some information placed in the supplementals give a more thorough idea on the extent of the work. They should therefore be placed in the main body (ED Fig 4A (+ add a legend on colours); ED fig 3A,B to be exchanged with Fig2A,B which is too complicated).

We have brought the information from the supplementals into the manuscript as requested. In particular ED Fig 3 was brought into the main manuscript as Figure 5. In addition, we have added a new figure (Figure 2) which has information on the seasonal variation in *Microsporidia MB* prevalence.

- If the authors suggest that blockade occurs before the oocyst stage, it would make sense to check whether melanized ookinetes can be observed in the mosquito guts. This could be done in the same experiment as oocyst counts mentioned above.

The suggestion that *Microsporidia MB* could cause an increase in melanization of ookinetes as an explanation for the *Plasmodium* protection phenotype is a very interesting one. We have not been able to directly investigate this as suggested but we have added to the manuscript an investigation of the interactions between *Microsporidia MB* and the host immune system from another angle. We carried out an RNA seq experiment on *Microsporidia MB* infected and *Microsporidia MB* uninfected midguts and ovaries. This experiment allowed us to identify some candidate genes for an RT-qPCR experiment that confirmed upregulation of two genes in *Microsporidia MB* infected midguts. While we understand that elucidating the precise mechanism of protection will require further study, these results are in line with the reviewer’s suggestions (immune priming). Altogether, they suggest that changes in host immunity and metabolism might make the vector ‘environment’ less permissive to the *Plasmodium* parasite.

Overall, the manuscript would require some polishing. This includes some typos (lines: 28, 31, 73, 101, 267), but also some work required on the text in general. The last sentence of the abstract should be tuned down not to oversell a tool that is not yet developed.

We have found and corrected the typos identified. The text has also been adjusted to fit the style of *Nature Communications*, with Introduction, Results, Discussion etc. We changed the last sentence of the abstract to represent our belief that these findings could have utility but not to suggest that a tool has been developed and is ready for use. It now reads, “As a *Plasmodium* transmission-blocking microbe that is non-virulent and vertically transmitted, *Microsporidia MB* could be investigated a strategy to limit malaria transmission”.

Finishing on a positive note, I would like to point that the authors have used a very relevant mode of infection with *Plasmodium*: they used the blood of patients, did not replace the sera to be in conditions as close as possible to what mosquitoes would really encounter in the field. Moreover, they used a good number of replicates.

We greatly appreciate this acknowledgement. We certainly faced many challenges in conducting these experiments in as close to natural conditions as possible; having *Microsporidia MB* infected mosquitoes ready for feeding AND finding a carrier required a lot of work and numerous failed attempts. However,

we think that having started with natural infection conditions and finding a protective effect makes our work more relevant and will help us to justify continuation of this work (some aspects of which could in the future be studied in *Plasmodium* infection models).

Reviewers' Comments:

Reviewer #3:

Remarks to the Author:

Line 149-152: The importance of this figure is not to see whether there is a significant positive correlation (which is misleading) between qPCR and oocyst count. It is to monitor whether a zero density is a zero oocyst. Indeed, we know that a single oocyst will be able to produce 100s-1000s sporozoites for transmission. Here, you have 7/20 false negatives and you conclude that "We confirmed that qPCR-based detection of oocysts had comparable accuracy and sensitivity to microscopy-based methods"? Am I reading it right? No way ! This implies that everything needs to be tuned down. It's not transmission blocking, even if it has a negative impact on vector competence. You would need to remove the prevalence data, ie Figure 4A, C; remove the correlation stats on the qPCR/oocyst chart; and really be more cautious about overselling transmission blocking. Transmission reducing is already interesting.

L.210: peritrophin

End of the abstract - investigated "as" a strategy to limit malaria transmission.

Sex ratio: put the width of each column depending on the sample size (e.g. proportional to log-transformed sample size) so that we can see easily that we should not emphasise too much the extremes which are n=2 and n=6. Average calculations are between replicates or overall sex ratio? As some replicates have only few mosquitoes, it might make sense to provide the overall sex ratio.

Responses to reviewer comments:

Reviewers' comments:

Reviewer #3 (Remarks to the Author):

Line 149-152: The importance of this figure is not to see whether there is a significant positive correlation (which is misleading) between qPCR and oocyst count. It is to monitor whether a zero density is a zero oocyst. Indeed, we know that a single oocyst will be able to produce 100s-1000s sporozoites for transmission. Here, you have 7/20 false negatives and you conclude that “We confirmed that qPCR-based detection of oocysts had comparable accuracy and sensitivity to microscopy-based methods”? Am I reading it right? No way ! This implies that everything needs to be tuned down. It’s not transmission blocking, even if it has a negative impact on vector competence. You would need to remove the prevalence data, ie Figure 4A, C; remove the correlation stats on the qPCR/oocyst chart; and really be more cautious about overselling transmission blocking. Transmission reducing is already interesting.

We understand that Supplementary Figure 4 was misleading, and we have endeavored to correct this. The goal was to demonstrate that qPCR and microscopy generate similar (although not identical) data for midgut oocyst infection rates and intensity. To avoid confusion, we have described the experiment and the results more clearly in the manuscript text. We have also removed the qPCR/oocyst chart and replaced it with a table that has actual values oocyst counts and qPCR intensities (Supplementary Table 1). Out of the 75 *Anopheles* midgut samples where we have data for both microscopy and qPCR, there were only two low density microscopy positive samples that were qPCR negative. The reviewer response indicated a higher number (7), which would suggest a much higher error rate (either in microscopy or qPCR). We believe that these two ‘positives’ are likely to be due to misidentification of oocysts, which is not altogether uncommon in the field laboratory setting. We have a high level of confidence in the technician that provided microscopy support, however, it is worth noting that our lab is much more experienced in molecular detection methods. Along these lines, we have conducted many experiments to confirm the validity of our qPCR-based assay, testing control *Plasmodium* from different sources in different forms and at a range of concentrations. In addition, each positive sample in this manuscript was re-tested by PCR and a number of them have also been sequenced. In light of this, we think it is far less likely that this discrepancy is related to issues in the detection of *Plasmodium* by the qPCR-based assay. We hope that by explaining these findings in the results and highlighting them clearly in the manuscript text and Supplementary Table 1, will enable readers to understand any potential limitations linked to this.

We agree that if our oocyst error rate had been very high (e.g. 7/20 as opposed to 2/21), it would have been pertinent to consider removing Figure 4C. However, in light of a much lower rate, we believe that it is reasonable to assume qPCR negative samples are indeed highly likely to be oocyst negative and therefore that 4C is a fair representation of our findings (as well as 4C, where an absence of sporozoites is demonstrated). We have also been more cautious to not say that oocyst formation is ‘blocked’. Since we also demonstrated the absence of sporozoites in *Microsporidia* MB-infected mosquitoes we retain the word ‘block’ in reference to sporozoites and overall transmission. It is also worth noting that ‘transmission blocking’ is commonly used in instances where absolute 100% blockage isn’t implied (e.g. the *Wolbachia*-Arbovirus field).

L.210: peritrophin

Corrected.

End of the abstract - investigated "as" a strategy to limit malaria transmission.

Corrected.

Sex ratio: put the width of each column depending on the sample size (e.g. proportional to log-transformed sample size) so that we can see easily that we should not emphasise too much the extremes which are $n=2$ and $n=6$. Average calculations are between replicates or overall sex ratio? As some replicates have only few mosquitoes, it might make sense to provide the overall sex ratio.

We agree with these suggestions and have made the changes.